# Synthesis and Characterization of Cyclodextrin-Based Polyhemiaminal Composites with Enhanced Thermal Stability

**DOI:** 10.3390/polym14081562

**Published:** 2022-04-11

**Authors:** Hoque Mohammed Jabedul, Mitsuo Toda, Nobuyuki Mase

**Affiliations:** 1Department of Optoelectronics and Nanostructure Science, Graduate School of Science and Technology, Shizuoka University, 3-5-1 Johoku, Hamamatsu 432-8561, Shizuoka, Japan; mase.nobuyuki@shizuoka.ac.jp; 2Department of Engineering, Graduate School of Integrated Science and Technology, Shizuoka University, 3-5-1 Johoku, Hamamatsu 432-8561, Shizuoka, Japan; 3Research Institute of Green Science and Technology, Shizuoka University, 3-5-1 Johoku, Hamamatsu 432-8561, Shizuoka, Japan

**Keywords:** polyhemiaminal, thermal stability, cyclodextrins, composite

## Abstract

Polyhemiaminal (PHA) polymers are a new class of thermosetting polymers that have recently gained attention owing to their high mechanical strength and excellent recycling behavior. However, low thermal stability is a common issue in PHA polymers due to the thermally labile crosslinked knots. Herein, crosslinked PHA polymer composites were synthesized by reacting formaldehyde with a precursor solution of 4,4ʹ-oxydianiline (ODA) and cyclodextrins (CDs) (α-, β-, and γ-). The material obtained under optimal conditions (ODA:CD molar ratio of 1:0.5, 37% aqueous solution of formaldehyde (formalin)) exhibited good film formability and high thermal stability with two characteristic decomposition phenomena and a high char yield. The early decomposition of CDs and char formation led to high thermal stability. Time-resolved NMR analysis was conducted to study hemiaminal bond formation via a condensation reaction between ODA and formaldehyde. Furthermore, PHA matrix formation was confirmed by the dissolution of the deposited CD layer in a solution of *N*-methyl-2-pyrrolidinone containing 8–9 wt.% LiBr at 80 °C and FTIR analysis. Based on the elemental analysis results, PHA network formation was confirmed by considering a single unit of the PHA network with CD composition, including the solvent and water.

## 1. Introduction

Crosslinked poly(hexahydrotriazine) (PHT) polymers exhibit high mechanical strength, chemical resistance, low density, and pH responsiveness and are consequently attracting increased attention for potential applications in adhesives, coatings, printing materials, and electronic sensors [1,2,3,4,5,6,7]. Polymers with triazene or hexahydrotriazine (HT) cores contain photosensitive and thermolabile crosslinks; therefore, they are often used as photopolymers in laser ablation applications [8,9]. Recently, the generation of PHT crosslinks via polyhemiaminal (PHA) intermediate formation at 50 °C followed by cyclization to PHT at 200 °C has been employed to prepare novel thermosetting polymers with an acid-triggered recyclability and good mechanical strength (Young’s moduli ~6.3 and ~14 GPa) exceeding those of known thermosets (Figure 1) [1]. Furthermore, the self-healing properties of organogels intended for adhesive material applications have been probed by computational and experimental model core analyses of supramolecular interaction-based dynamic covalent network formation in PHA [10]. Numerous thermosets have been prepared by copolymerizing reactive species, such as maleimide, epoxides, and acrylates, with PHT or PHA, and characterized in terms of mechanical properties and processability for large-scale applications [11]. Recently, interest has been growing in carbon-fiber-reinforced PHA composites owing to their recyclability, low cost, and ease of fabrication [12,13,14]. Although these materials exhibit high mechanical strength, recyclability, and self-healing properties, the presence of thermolabile PHT and PHA cores results in insufficient high-temperature performance [7,11]. Generally, thermal properties can be tuned by physically blending various thermoplastic resins or by incorporating reactive species into the reaction mixture to improve homogeneity through chemical modification [11]. In this context, CD/PHA composites were prepared via polycondensation of a precursor solution containing cyclodextrins (CDs) and 4,4′-oxydianiline (ODA) with formaldehyde in *N*-methyl-2-pyrrolidinone (NMP). These cyclic oligomers of glucose exhibit a unique thermal stability range (252–400 °C) and char-forming ability upon degradation [15] and are frequently used as environmentally benign diverse-functionality hosts for suitably sized hydrophobic guests in surfactant and functional polymer applications [16,17,18]. Although inclusion complexes are formed with hydrophobic guests in water, their implementation in thermosetting polymers is limited by factors such as solvent polarity, reactivity, and solid-state channel structure formation. CD dethreading occurs in polar solvents, whereas the reactivity of the complex is limited in solid-state channel inclusion complexes formed through strong hydrogen bonds [19,20]. In such cases, composite formation can be studied to improve the thermal stability of the PHA polymers. Bio-based natural fiber-reinforced composites are attractive alternatives to those based on glass and carbon fibers in automotive applications owing to their low cost and biodegradability [21,22]. Despite the potential of PHA and PHT crosslink-based polymers, their applications in bio-based natural fiber composites are scarce because of the lack of appropriate functionalization and solubilization procedures for characterization [6,10]. In particular, the synthesis of PHA or PHT-crosslinked polymers with CD composition has not yet been explored. To bridge this gap, we studied CD-based PHA composites and evaluated their thermal properties, solubilities, and recyclabilities. Among the three CDs (α-, β-, and γ-CDs), α-CD was preferentially evaluated because it is a readily available low-cost food-grade material. Hemiaminal bond formation before gelation was confirmed using real-time nuclear magnetic resonance (NMR) measurements. The solubility of the obtained polymers was evaluated using NMP/LiBr and dimethylacetamide (DMAc)/LiBr as solvents [23]. As CDs exhibit similar thermal characteristics, this study can be a relevant approach in understanding the formation of PHA-based cellulose composites. 

## 2. Materials and Methods

### 2.1. Materials

4,4’-oxydianiline (ODA), 4,4’-methylenedianiline (MDA), 2,2-bis(4-(4-aminophenoxy)phenyl)propane (BAPP), and *N*,*N*-dimethyl-1,4-phenylenediamine (DMPD) were purchased from Tokyo Chemical Industry Co. Ltd. (TCI) (Tokyo, Japan) and had purities of >98%. α-cyclodextrin (α-CD) (Food-grade), β-CD, γ-CD, *N*-methyl-2-pyrrolidinone (NMP), and formalin were purchased from Kanto Chemical Co. Inc. (Tokyo, Japan). Paraformaldehyde (PFA) powder (95%) was purchased from Nacalai Chemicals Ltd. (Tokyo, Japan).

### 2.2. Methods

Solid-state Fourier transform infrared FTIR (FT/IR-6300; power = 180 W; working range = 4000–700 cm^−1^; 64 scans; JASCO Corporation, Tokyo, Japan) spectra were recorded at 25 °C. Raman spectra were recorded on a laser spectrometer (NRS-7100; 32 scans, JASCO Corporation, Japan) featuring He-Cd and YAG laser sources with wavelengths of 325, 532, 785, and 1064 nm (600 mW) in the wavenumber range of 3200–100 cm^−1^. ^1^H (400.13 MHz) and ^13^C (100 MHz) NMR (Avance Ultrashield 400, Bruker Japan, Yokohama, Japan) spectra were recorded using DMSO-*d_6_* as the solvent and tetramethylsilane as an internal standard. Wide-angle X-ray diffraction (XRD) patterns of the powder and solid resin polymer samples were recorded under ambient conditions on a diffractometer (RINT-2200, Rigaku, Tokyo, Japan) equipped with a Cu K (λ = 1.54 Å) source. Differential scanning calorimetry (DSC) (DSC-60, Shimadzu, Japan) measurements were performed under Ar (25 mL/min) in two temperature ranges (−40 to 40 °C and 40 to 400 °C) at a heating rate of 10 °C/min. Thermogravimetric analysis-differential thermal analysis (TGA-DTA) (DTG-60A, Shimadzu, Kyoto, Japan) measurements were performed under Ar (25 mL/min) in the temperature range of 30–500 °C at a heating rate of 10 °C/min. Field-emission scanning electron microscopy (FE-SEM) imaging was performed on a JSM 6335F instrument (JEOL, Tokyo, Japan) using Au-coated samples to decrease the charge. Atomic force microscopy (AFM) was performed on an AFM VN-8010 instrument (Keyence, Osaka, Japan). Elemental analysis was performed using an elemental analyzer (EA1112, Thermo Electron, Yokohama, Japan). 

### 2.3. Characterization

To facilitate manipulation and comparative studies, both ODA-PHA (Path A, Figure 1 [1]) and CDs/ODA-PHA (Path B, Figure 2, a newly designed process) were synthesized. The reaction conditions are listed in Table 1, and the thermal characteristics and elemental analysis data are listed in Table 2.

#### 2.3.1. Synthesis of ODA-PHA

ODA-PHA was prepared as previously described [1]. Additionally, the respective condensation reactions were performed using formalin as a source of formaldehyde. 

#### 2.3.2. Synthesis of CD/ODA-PHA

A 100 mL oval-shaped round-bottom flask was charged with ODA (0.20 g, 1.02 mmol, 1.0 eq.), α-CD (0.50 g, 0.51 mmol, 0.5 eq.), and NMP (4.0 mL), and the mixture was sonicated in an ultrasonic bath for 60 min to obtain a clear solution (precursor solution), which was then treated with PFA or formalin (0.14 g, 4.60 mmol, 4.5 eq.). The flask was immersed in an oil bath equipped with a reflux condenser under Ar, and the temperature was slowly raised to 55–60 °C under stirring. Within 1 h, the mixture turned into a viscous liquid and an opaque gel was obtained after 3–4 h. The obtained product was cooled to 25 °C and mixed with acetone (4 mL). The precipitate was filtered, rinsed with acetone (6–8 mL), and washed with water (8–10 mL) to remove the excess α-CD. The resulting white waxy resin was vacuum-dried at 50 °C for 8–10 h. The product obtained at 0.83 g corresponded to ~90 wt.% (insoluble fraction), including the solvent trapped in the polymer network. 

### 2.4. NMR Spectroscopic Analysis

^1^H NMR (400 MHz, DMSO-*d_6_*) of ODA-PHA (Appendix A): δ (ppm) 6.73–6.53 (Ar–H), 5.90 (Ar–NH–), 4.72 (–CH_2_–), and 4.42 (–OH).

^13^C NMR (100 MHz, DMSO-*d_6_*) (Appendix A): δ (ppm) 149.1 (C_M_), 143.8 (C_N_), 119.4 (C_Q_), and 115.4 (C_P_), 89.9 (–CH_2_–), 84.5 (PFA).

^1^H NMR (400 MHz, DMSO-*d_6_*) of α-CD/ODA-PHA (Appendix A): δ (ppm) 6.97–6.45 (Ar–H), 6.25 (Ar–NH–), 5.56 (O_2_H), 5.52 (O_3_H), 4.82 (H_1_), 4.69 (–CH_2_–), 4.64 (O_6_H), 4.40 (–OH), 3.73–3.18 (H_5_ to H_2_).

^13^C NMR (100 MHz, DMSO-*d_6_*) (Appendix A): δ (ppm) 149.1 (C_M_), 144.2 (C_N_), 119.4 (C_P_), 115.6 (C_Q_), 102.4 (C_1_), 89.9 (–CH_2_–), 84.6 (PFA), 82.5 (C_2_), 73.7 (C_3_), 72.5 (C_4_ and C_5_), 60.5 (C_6_).

^1^H NMR (400 MHz, DMSO-*d_6_*) of β-CD/ODA-PHA (Appendix A): δ (ppm) 7.01–6.57 (Ar–H), 6.15 (Ar–NH–), 5.75 (O_2_H), 5.69 (O_3_H), 4.84 (H_1_), 4.68 (–CH_2_–), 4.49 (O_6_H), 4.40 (–OH), 3.64–3.25 (H_2_ to H_5_).

^13^C NMR (100 MHz, DMSO-*d_6_*) (Appendix A): δ (ppm) 149.2 (C_M_), 143.9 (C_N_), 119.4 (C_Q_), 115.7 (C_P_), 102.4 (C_1_), 89.9 (–CH_2_–), 84.5 (PFA), 82.0 (C_2_), 73.5 (C_3_), 72.9 (C_4_), 72.5 (C_5_), 60.4 (C_6_).

^1^H NMR (400 MHz, DMSO-*d_6_*) of γ-CD/ODA-PHA (Appendix A): δ (ppm) 6.69–6.55 (Ar–H), 6.21 (Ar–NH–), 5.79 (O_2_H), 5.76 (O_3_H), 4.89 (H_1_), 4.64 (–CH_2_–), 4.52 (O_6_H), 4.40 (–OH), 3.63–3.33 (H_2_ to H_5_).

^13^C NMR (100 MHz, DMSO-*d_6_*) (Appendix A): γ-CD: δ (ppm) 149.1 (C_M_), 144.2 (C_N_), 119.4 (C_Q_), 115.5 (C_P_), 102.1 (C_1_), 89.9 (–CH_2_–), 84.5 (PFA), 81.4 (C_2_), 73.4 (C_3_), 73.0 (C_4_), 72.6 (C_5_), 60.4 (C_6_).

### 2.5. FTIR Analysis

ODA-PHA (Figure 1): ν (cm^−1^) = 3373 (O–H), 2918 (C–H), 1678 (C=O of NMP) [24,25], 1497 (C=C) [26], and 1214 (C–N) [27]. 

α-CD/ODA-PHA: ν (cm^−1^) = 3347 (O–H), 2926 (C–H), 1663 (C=O of NMP), 1501 (C=C), 1232 (C–N), 1031 (C–O–C) [28].

### 2.6. Raman Analysis

ODA-PHA (Figure 2): ν (cm^−1^) = 3068 (N–H), 2928 (C–H), 1614 (C=C), 1434 (C–H), 1166 (C–C), and 925 (C–N) [29,30]. 

CDs/ODA-PHA: ν (cm^−1^) = 3070 (N–H), 2930 (C–H), 1613 (C=C), 1437 (C–H), 1168 (C–C), 930 cm^−1^ (C–N) [29].

## 3. Results and Discussion

Considering previous studies on PHA and PHT polymers (Figure 1), we studied PHA composite formation with three CDs (α -, β -, and γ-). In this respect, a sonicated precursor solution of monomers (ODA, MDA, and BAPP) was treated with PFA in NMP at 60 °C to obtain CD/PHA. Notably, NMP was reported to form a matrix with β-CD owing to the diffusion of water [31]. Both bulk polymerization and film formation were conducted to understand the matrix-assisted composite formation. The obtained CD/PHA composites were then subjected to thermal, solubility, and recyclability tests. The results showed that, in the presence of CD, aromatic diamines underwent polycondensation with formaldehyde to afford PHA composites. The incorporation of CD resulted in a marked change in the physical state, that is, the final products were obtained as solid resins as opposed to powders (Figure 2 and Appendix A). Different ODA:CD molar ratios (1:1 and 1:0.5) were used to optimize the properties of the resulting polymers, with the best result obtained at a ratio of 1:0.5. The condensation reactions were carried out according to Figure 2 (Path A) used to prepare the reference material [1] and (Path B) used to prepare CDs/PHA. The reaction was faster in the case of formalin (Table 1), indicating the promotional effect of water contained in this reagent, in agreement with a previous report [10]. Moreover, although formaldehyde had to be released from PFA by cracking, no such kinetic barrier was present in the case of formalin. The time required to accomplish condensation during heating at 60 °C was determined (based on gelation) to be 3–4 h for the α-CD-ODA precursor solution. Both ODA-PHA and CD/ODA-PHA films were prepared in formalin by heating on a Petri dish (Appendix A). The transparent α-CD-based film exhibited better flexibility than the reference ODA-PHA film did. Moreover, the ODA-PHA film shrank during thermal curing at 60 °C, whereas the α-CD/ODA-PHA films strongly adhered to the glass surface owing to their adhesive properties. In addition to the spectroscopic assessments, hemiaminal bond formation was confirmed by real-time time-resolved ^1^H NMR analysis at a probe temperature of 55 °C. Furthermore, elemental analysis confirmed the previously proposed formation of a solvent-stabilized PHA core [1] and suggested a component model analysis for both the ODA-PHA polymer and the CD/ODA-PHA composites (Appendix A). Finally, both the ODA-PHA polymer and CDs/ODA-PHA composites were recycled by decomposing in 1 N H_2_SO_4_ (aq.) and precipitating in 1 M Na_2_CO_3_ (aq.).

### 3.1. FTIR Analysis 

FTIR spectroscopy was used to analyze the composite formation. As ODA-PHA was formed via a polycondensation reaction, the intensity loss of the amine band (~3388 cm^−1^) and the emergence of the O–H band (~3388 cm^−1^) and C–H band (~2918 cm^−1^) were observed (Figure 1). Two characteristic bands at ~1497 cm^−1^ and ~1214 cm^−1^ were observed because of the C=C stretching of the phenyl group and the C–N stretching of either the amine part or the hemiaminal moiety. One extra band at ~959 cm^−1^ due to C–H bending was observed for ODA-PHA and did not stem from ODA, indicative of hemiaminal core formation. An extra band at ~1678 cm^–1^ observed even after vacuum drying at 60 °C for 8–12 h was ascribed to the C=O stretching of NMP, as previously reported for Path A synthesis [1]. By contrast, the CD/ODA-PHA composites showed combined bands for both the ODA-PHA core and CD. In addition to the characteristic C=C and C–N stretching bands, typical O–H stretching at ~3347 cm^−1^ and C–O–C stretching at ~1031 cm^−1^ were observed. Neither the characteristic amine bands nor an overlap of the O–H and N–H bands was observed for CD/ODA-PHA. 

### 3.2. Raman Analysis

In the Raman spectra of ODA-PHA and CD/ODA-PHA, the characteristic amine band of ODA (~3057 cm^−1^) disappeared (Figure 2), whereas a C–H band (~2928 cm^−1^) associated with hemiaminal bond formation appeared. Compared with ODA-PHA, an intense C–H signal (~2930 cm^−1^) was observed for CDs/ODA-PHA because of the overlap of the C–H bands of CD and hemiaminal bonds. A small sharp peak at ~3068 cm^−1^, which was observed for both polymers and did not stem from α-CD, was ascribed to N–H (secondary amine) stretching of the hemiaminal linkages. Furthermore, characteristic C=C stretching of the phenyl group at ~1614 cm^−1^ and C–N stretching of the hemiaminal linkage at ~925 cm^−1^ were observed for both ODA-PHA and CD/ODA-PHA [29,30]. Notably, the polymer composites exhibited identical but less intense Raman shifts than ODA-PHA within the range of 1700–500 cm^−1^. These results clearly confirmed the formation of CD/PHA composites. 

### 3.3. Powder XRD Analysis

No crystalline peaks related to ODA or the CDs were observed in the XRD patterns (Figure 3). The major peak of ODA-PHA appeared as a broad hump at ~17.9°, which was ascribed to the diffusion of scattered incident radiation [1]. By contrast, CD/ODA-PHA exhibited two major CD peaks at ~20.0° and ~20.1°. The significant shift in the major broad peak confirmed that the originally crystalline CD became amorphous upon composite formation [32]. Moreover, for the polymer composites containing α-, β-, and γ-CD, the signals corresponding to the ODA-PHA matrix shifted to 13.2°, 15.4°, and 16.2°, respectively, and were less intense than those of ODA-PHA. These results indicated that the polymer composites had an ODA-PHA matrix composed of CDs.

### 3.4. Hemiaminal Core Studies Using NMR Spectroscopy

Hemiaminal bond formation was monitored via real-time time-resolved ^1^H NMR analysis prior to gel formation at a probe temperature of 55 °C (direct procedure). ODA (0.02 g) was dissolved in DMSO-*d_6_*, and the NMR spectrum was recorded after 10 min. Another tube containing ODA (1.0 eq.), PFA (4.5 eq.), and DMSO-d_6_ was immediately analyzed at the starting point. Successive NMR spectra were recorded after each 10 min interval (Figure 4). The hemiaminal bond formation was observed after 10 min, as confirmed by the emergence of new peaks due to hemiaminal protons at 5.87 ppm (–NH–), 4.69 ppm (–OH), and 4.43 ppm (–CH_2_–), as well as consumption of PFA peak at 4.83 ppm (–CH_2_–). The relative integral values of the hemiaminal peaks increased, whereas those of the PFA peak increased with respect to time. The molar ratio was calculated based on the relative integral values of the hemiaminal peaks (–NH– and –OH) and PFA peaks (–CH_2_–), and plotted with respect to time. The molar ratio increased dramatically over time (Figure 4). ^1^H NMR measurements were performed using a model compound (DMPD). The emergence of hemiaminal peaks was observed at 5.30 ppm (–NH–), 4.54 ppm (–CH_2_–), and 4.39 ppm (–OH) (Appendix A). However, a synergistic methylene peak arising from the hemiaminal bond evolved near the methylene protons of the PFA. The integral values of the hemiaminal peaks increased with a decrease in PFA peaks. Composite formation was monitored via NMR measurements at 55 °C by reacting a precursor solution of α-CD, ODA, and formalin in DMSO-*d_6_*. As the reaction proceeded, some new peaks emerged at 5.98 ppm, 4.70 ppm, and 4.62 ppm due to hemiaminal protons, such as –NH–, –CH_2_–, and –OH, respectively (Appendix A). Notably, the aromatic and hydroxyl proton peaks of α-CD broadened significantly with time owing to gel formation. However, the α-CD cavity and anomeric protons remained unchanged for up to 120 min. 

Hemiaminal core formation was monitored indirectly. In this case, the required amounts of diamine and 100% formalin (4.5 eq.) were placed in a small flask, and DMSO-d_6_ was added instead of NMP. The mixture was then heated to 60 °C for 20 min, and NMR spectra were recorded. In the ^1^H NMR spectra, hemiaminal protons associated with (Ar–NH–), –CH_2_–, and –OH appeared at 5.90 ppm, 4.72 ppm, and 4.42 ppm, respectively (Appendix A). In the ^13^C NMR spectrum, a hemiaminal carbon peak appeared at 89.9 ppm (Appendix A). To monitor the composite formation, the required amounts of diamine and the CDs were dissolved in DMSO-d_6_, followed by sonication (1 h), the addition of formalin (4.5 eq.), and stirring. An aliquot was then transferred to an NMR tube and heated at 60 °C for 20 min. In the ^1^H NMR spectrum of α-CD/ODA-PHA, hemiaminal protons, such as (Ar–NH–), –CH_2_–, and –OH, were assigned at 6.25 ppm, 4.69 ppm, and 4.40 ppm, respectively. Notably, a hemiaminal carbon resonance peak was observed at the same position as ODA-PHA at 89.9 ppm for all composites.

### 3.5. TGA

In the TGA thermogram, ODA exhibited single-step weight loss (99.0%) associated with two sharp endothermic peaks at 198.6 °C and 346.3 °C (Figure 5 and Appendix A). Similarly, the thermogram of α-CD showed two major endothermic peaks, with a total weight loss of 99.2%. No characteristic endothermic peaks were observed for either ODA-PHA or CD/ODA-PHA. Typically, ODA-PHA showed two-step weight loss (30.4% and 28.8%) associated with endo- and exothermic peaks. The former was due to solvent and water volatilization, and the latter was related to network degradation. In contrast, thermograms of CD/ODA-PHA exhibited three-step weight loss: 23.3%, 12.6%, and 29.5% for α-CD/ODA-PHA; 16.7%, 15.3%, and 42.1% for β-CD/ODA-PHA; and 12.5%, 18.5%, and 38.0% for γ-CD/ODA-PHA (Figure 5). The first weight loss was attributed to solvent and water volatilization, whereas the second and third weight losses were attributed to CD decomposition and degradation of aromatic residues, respectively, leading to char formation (Appendix A). Notably, the final *T_d_* significantly increased upon composite formation with CD, which was ascribed to an insulating char barrier formed upon CD decomposition, which slowed the diffusion of oxygen and nitrogen (Appendix A) [15]. The acquired data were used to calculate the char yields at points corresponding to the decomposition of the CDs and the final decomposition of ODA-PHA and CD/ODA-PHA (Appendix A). All polymers had almost identical char yields at the final *T_d_*, with some variations observed only at the first *T_d_*. The obtained results indicated that the CDs acted as sources of primary char until a stable char was formed at the final *T_d_*. α-CD/MDA-PHA and α-CD/BAPP-PHA were further subjected to TGA to gain insight into the polymer decomposition behavior (Appendix A), which was found to depend on the type of central bond between the two aromatic units (Ar–O–Ar in ODA, Ar–CH_2_–Ar in MDA, and Ar–CMe_2_–Ar followed by two ether linkages in BAPP) and on the distance between the two crosslinks [33]. Although BAPP was expected to exhibit the highest bond strength among the three diamines, the presence of two ether linkages next to the aromatic carbons resulted in a bond strength slightly lower than that of MDA. The bond strength decreased in the order MDA > BAPP > ODA, which corresponds to the stability order determined by TGA for the three polymers.

### 3.6. DSC

In the DSC thermogram, ODA exhibited a sharp *T_m_* peak at 195 °C, which evolved into a broad amorphous peak upon polymerization (Appendix A). The *T_g_* of CD/ODA-PHA decreased in the order α- > β- > γ-CD, and that of ODA-PHA (107 °C) was similar to the previously reported values. Two major broad exothermic peaks were observed for ODA-PHA and CDs/ODA-PHA. The first exothermic peak and the corresponding enthalpy were determined to be 258 °C and 8.84 J/g for ODA-PHA, 247 °C and 119.8 J/g for α-CD/ODA-PHA, 241 °C and 74.8 J/g for β-CD/ODA-PHA, and 237 °C and 59.2 J/g for γ-CD/ODA-PHA, respectively. Clearly, the incorporation of CD increased the enthalpy of the first exothermic peak [34]. The enthalpy related to the second exothermic peak at the final decomposition point was determined to be 80.8 J/g for ODA-PHA, 67.4 J/g for α-CD/ODA-PHA, 106.3 J/g for β-CD/ODA-PHA, and 94.5 J/g for γ-CD/ODA-PHA. DSC analysis conducted from −40 to 40 °C revealed one small transition for ODA-PHA (at −4.4 °C) and α-CD/ODA-PHA (at −7.1 °C). These peaks, ascribed to the *T_g_* of these polymers in the low-temperature region, have not been reported previously (Appendix A). The emergence of exothermic peaks and a shift in the final *T_d_* value indicated that the CDs improved the thermal stability of the CD/ODA-PHA polymer composites. 

### 3.7. Elemental Analysis

Based on the experimental elemental composition data, a hypothetical model was proposed by considering a single unit of the PHA network with the solvent, trapped water, and CD (α-, β-, or γ-) compositions (Appendix A). The results of the elemental analysis conducted for ODA-PHA and CD/ODA-PHA prepared at ODA:CD molar ratios of 1:1 and 1:0.5 were consistent with the theoretical values determined by considering hypothetical model components (Table 2). Similarly, good agreement was observed between the experimental results and theoretical values for α-CD/MDA-PHA; however, a slight deviation in the experimental composition was observed for α-CD/BAPP-PHA, which was ascribed to the different compositions of CD distributed in the composite (Appendix A). The elemental composition was calculated using the following components: (X_1_) ODA, (X_2_) hemiaminal, (X_3_) NMP, (Y) water, and (Z) CD.

### 3.8. FE-SEM

FE-SEM images of the ODA-PHA sample revealed a 3D porous structure [35], whereas those of α-CD/ODA-PHA showed layered structures with small grooves due to the aggregation of α-CD in the polymer matrix. (Figure 6). 

### 3.9. AFM

AFM analysis revealed a smooth surface for the α-CD/ODA-PHA film, whereas a rougher surface was observed for the ODA-PHA film (Figure 7). The surface smoothness could be attributed to a layer of α-CD composited on the ODA-PHA rough surface in α-CD/ODA-PHA.

### 3.10. Solubility Assay of ODA-PHA and CD/ODA-PHA in NMP/LiBr

DMAc/LiCl and DMAc/LiBr with salt contents of 8–9 wt.% were used as alternative solvents to solubilize polyrotaxanes and facilitate their acetylation and dansylation [23]. The low solubility of polyrotaxanes is due to intra- and intermolecular hydrogen bonding between hydroxyl groups [17,23]. However, these solvents destroy the hydrogen bonds through ionization. In this study, NMP/LiBr and DMAc/LiBr were employed to evaluate the solubilities of ODA-PHA and CDs/ODA-PHA. The maximum proportion of CD/ODA-PHA films (<68 wt.% of initial weight) was dissolved in the NMP/LiBr mixture (8–9 wt.% LiBr) at 80–90 °C in 3–5 h, and the remaining undissolved film was collected for FTIR analysis. By contrast, the ODA-PHA film remained unaffected at this temperature even when the treatment was continued for >72 h. A soluble portion of CD/ODA-PHA films was precipitated in water, filtered, washed with methanol, and vacuum-dried for 24 h. Notably, its FTIR spectrum featured all characteristic bands of α-CD/ODA-PHA, indicating oligomers of ODA-PHA and α-CD (Appendix A). The undissolved portion of the α-CD/ODA-PHA film was washed with water and methanol several times, kept in methanol for 24 h, and vacuum-dried at 60 °C for 3–4 h. The FTIR spectra of the undissolved portion showed characteristic hemiaminal bands such as 2913, 1498, and 1202 cm^−1^ for C–H, C=C, and C–N stretching, respectively, confirming the formation of the ODA-PHA matrix (Appendix A).

### 3.11. Diamine Recovery Test

A diamine recovery test was performed by digesting the α-CD/ODA-PHA composite in 1 N H_2_SO_4_, followed by precipitation in 1 M Na_2_CO_3_. Approximately 0.2 g of the composite was placed in a vial, to which 1 N H_2_SO_4_ (6 mL) was added. The polymer was disintegrated within 5 min to obtain a clear solution. When 1 M Na_2_CO_3_ solution was added dropwise, CO_2_ bubbling was observed, along with an increase in pH and the appearance of precipitates. After maintaining a pH of approximately 7–8, precipitation was complete, and bubbling was stopped. The precipitates were filtered and washed with excess hot water to remove CD, which was collected after vacuum drying. The FTIR and NMR spectra of the precipitate were recorded and compared with those of the pristine ODA (Appendix A). In the FTIR spectrum, the characteristic C=C stretching of the phenyl group (~1696 cm^−1^) and C–N stretching (~1214 cm^−1^) were observed. The characteristic peaks of the hemiaminal protons were absent in the ^1^H NMR spectrum, indicating ODA recovery.

## 4. Conclusions

In this study, the formation of CD/PHA composites was investigated, revealing that the incorporation of CD resulted in high thermal stability and the emergence of two decomposition phenomena. The early-stage decomposition of the CDs formed a char barrier that improved the thermal stability of the PHA composites. The adopted catalyst-free simple synthesis approach employed readily available chemicals such as food-grade α-CD, formalin, and a simple aromatic diamine, and is, therefore, suitable for industrial applications. The introduction of CDs through high-throughput synthesis may potentially reduce material costs, and the nontoxicity and environmental friendliness of CDs coupled with the acid-triggered recyclability of PHA polymers make the developed process environmentally viable. The early-stage decomposition of the CDs combined with the thermolabile PHA knot makes the prepared compound a potential component of flexographic materials for laser engraving. Previous laser ablation studies employed various aliphatic polyether diamine-based PHT polymers with 10–20 wt.% carbon black to obtain flexible polymers [3]. The CD/PHA composites prepared herein exhibited inherent flexibility and could act as a carbon source for laser ablation studies. Finally, this study can also be related to the formation of PHA-based cellulosic composites.

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
