# Peer review of "Synthesis and Characterization of Cyclodextrin-Based Polyhemiaminal Composites with Enhanced Thermal Stability"

_polymers, 2022, doi:10.3390/polym14081562_

Round 1
Reviewer 1 Report
Dear Editor
Through this work, the author's goal is to synthesize crosslinked PHA polymer composites by reacting formaldehyde (formalin) with a precursor solution of 4,4ʹ-oxydianiline (ODA) and cyclodextrins (CDs) (α-, β-, 17 and γ-) to overcome the low thermal stability characterization of PHA polymers, due to the crosslinked knots.
They have discussed the material obtained under optimal conditions (ODA: CD molar ratio of 1:0.5, 37% aqueous solution of formaldehyde (formalin) and confirmed that they exhibited good film formability and high thermal stability with two characteristic decomposition phenomena and a high char yield.
The purpose has been presented in a clear way and does not need any addition. The work is suitable for publication in your respectable journal.
Thanks
Author Response
General Concern:
Through this work, the author's goal is to synthesize crosslinked PHA polymer composites by reacting formaldehyde (formalin) with a precursor solution of 4,4ʹ- oxydianiline (ODA) and cyclodextrins (CDs) (α-, β-, 17 and γ-) to overcome the low thermal stability characterization of PHA polymers, due to the crosslinked knots.
They have discussed the material obtained under optimal conditions (ODA: CD molar ratio of 1:0.5, 37% aqueous solution of formaldehyde (formalin) and confirmed that they exhibited good film formability and high thermal stability with two characteristic decomposition phenomena and a high char yield.
The purpose has been presented in a clear way and does not need any addition. The work is suitable for publication in your respectable journal.
Author response:
It is our great pleasure to thank the reviewer for his great appreciation.

Reviewer 2 Report
The work reported in the paper is quite innovative and interesting. This research paper is reviewed for publication in Polymers-MDPI Journal. The work can be accepted for publication following minor revision. Few points are –
[1] Please provide crosslinking density and thermodynamics of the composites.
[2] Please provide abbreviation section in the paper.
[3] In materials section, please provide commercial name of the materials used in present work.
[4] Page 5, #line 193, where is Table S1? #line 195, where is figure S3 and S4?
[5] In Table 2, how authors calculated the C/H/N/O content?
[6] In section 3.9, AFM images are missing? Please provide them?
Author Response
General Concern:
The work reported in the paper is quite innovative and interesting. This research paper is reviewed for publication in Polymers-MDPI Journal. The work can be accepted for publication following minor revision. A few points are –
[1] Please provide crosslinking density and thermodynamics of the composites.
[2] Please provide abbreviation section in the paper.
[3] In materials section, please provide commercial name of the materials used in present work.
[4] Page 5, #line 193, where is Table S1? #line 195, where is figure S3 and S4?
[5] In Table 2, how authors calculated the C/H/N/O content?
[6] In section 3.9, AFM images are missing? Please provide them?
Author response: Many thanks to the reviewer for scrutinizing our manuscript and for his valuable comments.
Concern # 1:
Please provide crosslinking density and thermodynamics of the composites.
Author response: By using real-time NMR assay, we have confirmed the hemiaminal cross-link formation with respect to time by the condensation reaction of diamine and paraformaldehyde or formalin (Figure 4). In this figure time versus molar ratio plot reveals the continuous molar ratio changes. On the other hand, a composite NMR assay at 55 °C also reveals the formation of hemiaminal cross-link (Figure S1j, supporting information). However, due to time constrain, we were not unable to measure the cross-link density. We have taken this issue seriously for further improvement of our research by swelling experiments or other methods.
Author action: Page 8, line 241–253, and line 256–263; page 7, Figure S1j (supporting information).
Concern # 2:
Please provide abbreviation section in the paper.
Author response: Abbreviations are defined at the first use in the manuscript. However, we could not find any section to put separate abbreviations. If it is possible to put an extra section, we will be obliged to put an abbreviation section.
Concern # 3:
In materials section, please provide commercial name of the materials used in present work.
Author response: Commercial names of the materials are provided in the materials section.
Author action: Page 2, lines 73–78.
Concern # 4:
Page 5, #line 193, where is Table S1? #line 195, where is figure S3 and S4?
Author response: As mentioned on page 5, line 193 about Table S1. This table is in the supporting information. As polymer and composite recycling figures are separately discussed in section 3.11 (Diamine recovery test), We have modified the sentence and removed figure S3 and S4 citations.
Author action: Page. 8, Table S1 (supporting information); page. 13, lines 391–393 (modified); page 5, lines 184–186 (removed figure S3 and S4 citation).
Concern # 5:
In Table 2, how authors calculated the C/H/N/O content?
Author response: Based on experimental elemental analysis data obtained from the elemental analyzer, we have represented a hypothetical model by considering four components including a single unit of hemiaminal core (C1), NMP solvent (C2), cyclodextrin (C3), and water (C4) (Figure S2 in supporting information). Further, the theoretical elemental composition percentage was calculated by using an empirical formula represented in supporting information on page 8, (Figure S2)
Author action: Page 8, Figure S2 in the details caption (supporting information).
Concern # 6:
In section 3.9, AFM images are missing? Please provide them?
Author response: Previously AFM image was in the supporting information. Now, we have included the AFM image in the manuscript (Figure 7) and removed it from its previous position in supporting information (Figure S6)
Author action: Page 12, Figure 7, line 356.
